# Improving turbulent airflow direction measurements for fiber-optic distributed sensing using numerical simulations

Mohammad Abdoli<sup>1,2</sup>, Reza Pirkhoshghiyafeh<sup>3</sup>, and Christoph K. Thomas<sup>1,4</sup>

Correspondence: Mohammad Abdoli (mohammad.abdoli@geo.rwth-aachen.de)

Abstract. This study investigates the impact of microstructure geometry on the thermal and turbulence responses of electrically heated fiber-optic (FO) cables under varying flow conditions and turbulence intensities for the purposes of sensing flow direction. The underlying measurement principle is the directionally sensitive heat loss from electrically heated FO cables with imprinted microstructures exposed to turbulent airflows resembling a long hot-wire anemometer. Using the COMSOL Multiphysics 6.0 finite-element software, this study explores a wider range of different configurations of filled-coned and hollow-coned microstructures of varying size compared to existing studies. The research identifies optimal combinations which maximize temperature differences ( $\Delta T$ ) across FO cables with cones pointing in opposite directions while balancing key design criteria such as sensitivity to wind speed and minimizing the FO-cables' PVC coverage. We demonstrate that FO cables with hollow-coned microstructures (radius = 24 mm, height = 24 mm, and spacing = 15 mm) outperform their filled-coned counterparts, maintaining  $\Delta T$  values above 2 K across a broader range of wind speeds and turbulence intensities. Notably, the hollow-cone configuration sustains a temperature difference of up to 0.8 K at a 60° wind attack angle. The findings implicate substantial improvements for an optimized FO cable design in atmospheric boundary layer studies, enabling more accurate measurements of wind direction, distributed turbulent heat fluxes, and vertical wind speed perturbations using fiber-optic distributed sensing (FODS). Future work shall validate the findings under field conditions to assess the robustness and real-world applicability of the optimized design.

**Keywords.** Fiber-Optic Distributed Sensing (FODS), Microstructure Approach, Turbulence Measurement, Computational Modeling, COMSOL Multiphysics

<sup>&</sup>lt;sup>1</sup>Micrometeorology Group, University of Bayreuth, Universitätsstraße, 95447 Bayreuth, Bayaria, Germany

<sup>&</sup>lt;sup>2</sup>Physical Geography and Climatology, RWTH Aachen University, Wüllnerstraße 5b, 52062 Aachen, North Rhine-Westphalia, Germany

<sup>&</sup>lt;sup>3</sup>School of Mechanical Engineering, University of Tabriz, Bahman Boulevard 29, 5165665931 Tabriz, Iran

<sup>&</sup>lt;sup>4</sup>Bayreuth Center for Ecology and Environmental Research (BayCEER), University of Bayreuth, Universitätsstraße, 95447 Bayreuth, Bayaria, Germany

# **Abbreviations**

| Parameter                        | Symbol               | Description                                                             | Units                             |
|----------------------------------|----------------------|-------------------------------------------------------------------------|-----------------------------------|
| Temperature                      | T                    | Absolute temperature                                                    | K                                 |
| Turbulent kinetic energy         | K                    | Turbulent kinetic energy                                                | $\mathrm{m^2 \cdot s^{-2}}$       |
| Inlet velocity                   | $U_{in}$             | Velocity of the airflow at the inlet                                    | $\mathrm{m}\cdot\mathrm{s}^{-1}$  |
| Radius                           | r                    | Radius of the filled-coned or hollow-coned microstructure               | mm                                |
| Length                           | L                    | Length of the Channel                                                   | mm                                |
| Height                           | h                    | Height of the microstructure                                            | mm                                |
| Spacing                          | s                    | Spacing between microstructures                                         | mm                                |
| Wind attack angle                | θ                    | Wind attack angle                                                       | degrees                           |
| Turbulence intensity             | TI                   | Intensity of turbulence in the flow                                     | Dimensionless                     |
| Turbulence integral length scale | TL                   | Integral length scale of turbulence                                     | m                                 |
| Interface temperature            | $T_0$                | Temperature at PVC and fiber interface                                  | $^{\circ}\mathrm{C}$              |
| Solid density                    | $ ho_s$              | Density of the solid material                                           | $\mathrm{kg}\cdot\mathrm{m}^{-3}$ |
| Specific heat                    | $C_p$                | Specific heat capacity at constant pressure                             | $J \cdot kg^{-1} \cdot K^{-1}$    |
| Translational velocity           | $\mathbf{u}_{trans}$ | Velocity vector of translational motion                                 | $\mathrm{m}\cdot\mathrm{s}^{-1}$  |
| Heat flux (conduction)           | q                    | Heat flux by conduction                                                 | $W \cdot m^{-2}$                  |
| Heat flux (radiation)            | $\mathbf{q}_r$       | Heat flux by radiation                                                  | $\mathrm{W}\cdot\mathrm{m}^{-2}$  |
| Thermal expansion                | $\alpha, \alpha_p$   | Coefficient of thermal expansion                                        | $K^{-1}$                          |
| Stress tensor                    | S                    | Second Piola-Kirchhoff stress tensor                                    | Pa                                |
| Heat source                      | $Q_{add}$            | Additional heat sources                                                 | $W \cdot m^{-3}$                  |
| Fluid density                    | ρ                    | Density of the fluid                                                    | $\mathrm{kg}\cdot\mathrm{m}^{-3}$ |
| Fluid pressure                   | p                    | Pressure in the fluid                                                   | Pa                                |
| Viscous stress tensor            | au                   | Stress tensor representing viscous forces                               | Pa                                |
| Eddy viscosity                   | $\mu_T$              | Turbulent eddy viscosity                                                | Pa·s                              |
| Turbulent dissipation            | $\epsilon$           | Rate of turbulent kinetic energy dissipation                            | $\mathrm{m^2 \cdot s^{-3}}$       |
| Production term                  | $P_k$                | Production term for turbulent kinetic energy                            | $\mathrm{m^2 \cdot s^{-3}}$       |
| Constant for turbulence          | $C_{\mu}$            | Constant in $k-\epsilon$ model ( $C_{\mu}=0.09$ )                       | Dimensionless                     |
| Turbulence constant 1            | $C_{\epsilon 1}$     | Constant in $\epsilon$ equation ( $C_{\epsilon 1} = 1.44$ )             | Dimensionless                     |
| Turbulence constant 2            | $C_{\epsilon 2}$     | Constant in $\epsilon$ equation ( $C_{\epsilon 2}=1.92$ )               | Dimensionless                     |
| Turbulence constant 3            | $\sigma_k$           | Constant for turbulent kinetic energy equation ( $\sigma_k = 1.0$ )     | Dimensionless                     |
| Turbulence constant 4            | $\sigma_{\epsilon}$  | Constant for turbulent dissipation equation ( $\sigma_{\epsilon}=1.3$ ) | Dimensionless                     |

Table 1. Nomenclature

## 1 Introduction

Distributed temperature sensing (DTS) uses fiber-optic (FO) cable to measure temperature in continuous, defined sections along FO cables. DTS measurements involve sending a laser pulse into the FO cable, where parts of it are reflected. While most optical energy undergoes elastic scattering maintaining the original wavelength, some is absorbed and re-emitted at different wavelengths, known as inelastic Raman backscatter. The longer wavelength backscatter, called Stokes, has a nearly temperature-independent amplitude. The shorter wavelength backscatter, known as Anti-Stokes, has an amplitude which varies with temperature. By measuring the natural logarithm of the ratio of Stokes to Anti-Stokes backscatter averaged over continuous sections of the FO cable by means of range-gating, the temperature along the entire cable can be spatially resolved (Ukil et al., 2012; Selker et al., 2006; Thomas and Selker, 2021). Since their development in the 1980s, DTS technology has advanced significantly, enabling continuous temperature measurements with a resolution of 0.01°C and spatial resolutions as fine as a few meters or even decimeters. These sensors have been widely used in various applications, including dam structural health monitoring (SHM) (Bado, 2021), pipelines, tunnels (Ishii et al., 1997), offshore oil and gas installations (Nakstad and Kringlebotn, 2008; Johny et al., 2021), mines (Silva et al., 2022), and also in research areas such as hydrological investigations(Selker et al., 2006; Tyler et al., 2009; Bense et al., 2016; Taniguchi, 1993), soil moisture measurements (Steele-Dunne et al., 2010), and atmospheric measurements of turbulence and air temperature (Thomas et al., 2012; Keller et al., 2011; Peltola et al., 2021; des Tombe et al., 2020; Higgins et al., 2018; de Jong et al., 2015; Fritz et al., 2021). Using fiber-optic distributed sensing (FODS) in atmospheric measurement has gained more attention in the study of the stable boundary layer, particularly because the turbulence dynamics in stable and very stable boundary layers are only insufficiently captured by conventional point-based sensors in combination with theoretical frameworks such as the Monin-Obukhov similarity theory, the Kolmogorov spectrum, and Taylor's hypothesis of frozen turbulence (Sun et al., 2012). High-resolution FODS has emerged as a powerful tool for probing the stable boundary layer and unraveling its underlying physical processes since it enables the simultaneous spatiotemporal observation of key atmospheric parameters such as temperature, wind speed, and wind direction. In this context, Zeeman et al. (2015) explored the thermal structure of near-surface motions in the nocturnal boundary layer using FODS, revealing detailed temperature structures under stable conditions, including gradients and intermittent patterns at the meter scale. Huss and Thomas (2024) examined the efficiency of vertical heat transport and the coupling mechanisms governing turbulence in the stable boundary layer (SBL). Utilizing data from sonic anemometers and FODS, the study identified appropriate modeling approaches for quantifying turbulent surface heat flux under stable conditions, Furthermore, Pfister et al. (2021a, b) detected semi-stationary thermal submesofronts in the nocturnal boundary layer using FODS, while Mack et al. (2025) utilized airborne FODS during the Arctic polar night to capture key features of the SBL, which were then compared against the HARMONIE-AROME model.

The development of wind speed measurement method using FODS (Sayde et al., 2015; van Ramshorst et al., 2020) has opened new insights into the application of FODS in continuous wind speed measurement, suggesting that this method could be used to spatially resolve turbulence. In 2019, the Darkmix project aimed at developing a 3D large eddy observation (LEO) technique using DTS with the goal of investigate the weak-wind and stable boundary layer across different land uses, including

grassland, forest, and urban areas (Lapo et al., 2022). Within this project, Lapo et al. (2020) developed a method known as the "microstructure approach," which uses Computational Fluid Dynamics (CFD) modeling in concert with wind tunnel experiments to determine turbulent wind direction and speed using actively heated FO cables with imprinted filled-coned microstructures. This method employs the difference in convective heat loss from the filled-coned fibers when the wind flows along the direction of the cones compared to a cable with cones pointing in the opposite direction.

The principle behind using microstructures for sensing the wind speed and wind direction builds on convective heat transfer theory and is conceptually related to hot-wire anemometry (Sayde et al., 2015; van Ramshorst et al., 2020). When the fiber is heated above ambient air temperature, the cooling rate depends on the turbulent heat flux from the cable surface to the air, which scales with wind speed. By introducing asymmetric microstructures onto paired cables, oriented in opposite directions, the effective aerodynamic roughness of the surfaces is altered. This geometric asymmetry modifies the convective heat loss experienced by each cable. According to Owen and Thomson (1963), the convective heat flux from a roughned surface depends on air density, the specific heat of air, air–surface temperature difference, friction velocity, the effective roughness height, kinematic viscosity, and the Prandtl number. The orientation of the microstructures directly affects the effective roughness: cones aligned with the flow increase roughness, enhancing turbulent mixing and convective cooling, while cones pointing into the flow reduce roughness and suppress heat loss. This asymmetry in convective heat transfer produces a measurable temperature difference between paired cables. The sign of this temperature difference provides a robust signal for determining wind direction, and the magnitude of the temperature difference correlates with the wind speed.

70

This method was successfully tested in a field experiment by Freundorfer et al. (2021), where the wind direction was calculated using FODS and the accuracy of  $\leq 15^{\circ}$ . Subsequently, Abdoli et al. (2023) attempted to compute continuous turbulent vertical airflow and air temperature perturbations to compute the spatially resolved within a forest subcanopy using FODS and the microstructure approach and compared it with eddy covariance observations. In this study, the vertical wind speed was calculated using the differential heat loss approach described above, which demonstrated the potential of the microstructure approach to not only observe the wind direction but also measure the wind speed along the fiber. Nevertheless, this was only effective in weak-wind conditions with speeds typically less than 2 ms<sup>-1</sup>. One significant limitation of this methodology was the geometry of the microstructures printed onto the cable characterized by a high sensitivity of the vertically pointing heated and filled-coned FO cables to lateral wind speeds orthogonal to the cable, which resulted in the method becoming ineffective at horizontal wind speeds exceeding 0.2 ms<sup>-1</sup>. Despite these limitations, the effort demonstrated the considerable potential of the microstructure approach to observe turbulent airflow. However, further refinements are necessary to enhance its accuracy and performance in real-world turbulent flows. Furthermore, in the proof-of-concept study on the microstructure approach (Lapo et al., 2020), the selection of microstructure form, shape, size, and spacing was based on a limited set of combinations. The selection was made exclusively on the basis of modeling the turbulence around the microstructures, and not modeling the heat transfer explicitly. We here test a much broader range of microstructure forms, shapes, sizes, and spacing combinations beyond those tested by Lapo et al. (2020) with incorporated heat exchange simulation added to the model bears the potential to improve the utility of this approach in sensing the turbulent flow direction.

In this numerical study, we aim at overcoming the major weaknesses of the existing microstructure approach to enhance wind speed and direction measurements using filled-coned FO cables while minimizing their sensitivity to lateral wind flows. The microstructures considered include filled-coned (similar to the one used in Lapo et al. (2020)) for comparison with the previous numerical study and hollow-coned. The hollow-coned microstructure is selected to decrease the fiber's sensitivity to lateral airflows and reduce the fiber length covered with microstructures, which in turn increases its sensitivity to the heat loss and hence temperature difference. The tested microstructures vary in combinations of radius, height, and spacing, with a focus on turbulence and conjugate heat transfer. Our primary objective is to improve the temperature difference between the forward (wind directed towards the cone apex) and backward (wind opposite to the cone apex) coned fibers. A larger temperature difference between the forward and backward fibers ensures that the signal exceeds the reported measurement error 0 to 0.1 K reported by (Schilperoort et al., 2020). It therefore enhances the precision of the microstructure approach at higher wind speed ranges and increasing the persistence of the temperature difference under lateral wind conditions. Additionally, we aim at examining the sensitivity of the temperature difference for the most promising configurations to turbulence intensity (*TI*) and wind attack angle to increase the microstructure's utility for real-world turbulence flow sensing applications characterized by a large degree of space and time variability in magnitude, direction, and strength.

#### 2 Materials and Methods




This study employed the finite-element software COMSOL Multiphysics 6.0 to simulate heat transfer and fluid flow around two distinct microstructure designs, a filled-coned and a hollow-coned structure attached to a thin fiber optic (FO) cable. The use of COMSOL for coupled heat transfer and fluid-solid interaction problems has been widely validated in similar studies (Chen et al., 2023a; Sogukpinar, 2020; Sun et al., 2024; Ma et al., 2025; Kalantari et al., 2021). In particular, Chen et al. (2023a) and Kalantari et al. (2021) specifically employed the  $K-\epsilon$  turbulence model to heat-transfer simulations focusing on the thermal environment of a high geothermal tunnel and the heat transfer of fin-and-tube heat exchangers, respectively, thereby reinforcing the suitability of our selected model for the present study.

The modeling was conducted using incompressible air, with the study type set to stationary. All computations were executed on a desktop computer equipped with an Intel(R) Core(TM) i7-8700 CPU (3.20 GHz) and 64 GB of RAM. Fig. 1 illustrates the schematic of the filled-coned microstructures attached to a FO cables. The computational domain was determined parametrically to locate six microstructures in length. The height was determined to be 16r, with r being the radius of the cone. The separation distance between the forward and backward FO cables was determined to be 8r. The dimensions were selected and optimized with several test runs to maintain accuracy and reduce computational costs. Solid copper was used as the FO cable material, and polyvinyl chloride (PVC) was chosen for the microstructures, and gas form air was used for the rest volume of the model. It is important to note that FO cables currently used in atmospheric measurements typically consist of a stainless steel sheath filled with gel, encased in a polyethylene (PE) layer which has about a 1s effective response time (Thomas et al., 2012). In this study, solid copper was used as a substitute to simplify the model and focus on the geometry of the microstruc-

tures rather than the properties of FO-cable. These materials were selected from COMSOL's built-in library, which includes all standard properties such as dynamic viscosity, heat capacity at constant pressure, density, and thermal conductivity.





The K- $\epsilon$  turbulence model was used to simulate the turbulence while the 'Heat Transfer in Solids and Fluids' module was employed for modeling the heat transfer between the conical fiber and the surrounding air. The K- $\epsilon$  model, first introduced by Launder and Sharma (1974), is one of the most widely applied turbulence models. It solves two transport equations for the turbulent kinetic energy (K) and its dissipation rate ( $\epsilon$ ), from which the turbulent viscosity is derived. While the model is numerically robust and computationally efficient for fully developed turbulent shear flows, its formulation is based on the assumption of isotropic turbulence and employs empirical constants as well as wall functions. These limitations reduce its accuracy in complex flow conditions, particularly in the presence of adverse pressure gradients, rotation, or separation (Puntigam et al., 2025). Nevertheless, due to the widespread application of this model in turbulence research and to allow comparability with the study of Lapo et al. (2020), we chose to employ the K- $\epsilon$  model in this study.

The inlet wind  $(U_{in})$  was applied from left to right as shown in Fig. 1a with the turbulence intensity (TI) set to 0.05, which corresponds to moderate turbulence in the atmospheric boundary layer (Watkins, 2012). The TI is defined as  $\frac{\sigma_U}{\pi}$ , where  $\sigma_U$ represents the standard deviation of the wind over an average period, and  $\overline{U}$  is the total wind speed averaged over the same time period. Fig 1b shows a conceptual schematic of the temperature along the fiber for both forward and backward coned fibers, along with the temperature difference between them. This temperature difference arises from the cone orientation. In forward coned fibers, the flow penetrates the cone openings and enhances cooling of the fiber. In contrast, in backward coned fibers, the cone acts as a shelter that reduces exposure to the flow and thereby lowers the cooling rate. This mechanism underpins the microstructure approach, which generates a directional temperature difference between forward and backward coned fibers. The turbulence length scale is set to built-in function of COMSOL as "based on geometry". To test the sensitivity of the model to TI, a range of 0.01 to 0.4 was applied to the model (Table 2). All boundaries except the inlet and outlet were set to act as a wall. The ambient air temperature of the model was set to 20 °C and the FO cable temperature was set to 45 °C to create a 25K temperature difference between the FO cable and the ambient air. This temperature difference was selected to align with the range used in previous field experiments (Lapo et al., 2022), where FO cables were heated at a rate of 4.5 Wm<sup>-1</sup>, inducing temperature differences between 4.2 K and 31.0 K. In addition, the temperature difference between air and FO cable was chosen to be large enough to prevent excessive cooling within the wind speed range of the modeling. Based on the initial model runs, the inlet wind speed range was varied between 0 and 4.0 ms<sup>-1</sup> at 0.5 ms<sup>-1</sup> steps between 0 and 1 ms<sup>-1</sup>, and 1  $ms^{-1}$  steps for higher wind speeds (Table 2).

A physics-controlled mesh, generated by COMSOL Multiphysics to align with the specific settings of the model's physics interface, was employed with a standard element size, as illustrated in Fig. 2. Additional adjustments were applied to the mesh manually to ensure adequate resolution of the boundary-layer profiles and the microstructure edges. Our design ensured that the mesh geometries for the two-dimensional (2D) and three-dimensional (3D) simulations matched perfectly. The first part of the study was done with 2D modeling of 64 different geometry combinations since the model was symmetric and the inlet wind was aligned with the fibers. The 2D modeling is also used to analyze the effects of different turbulence intensity ranges.

| Parameters                   | Range                     |
|------------------------------|---------------------------|
| $U_{in} (\mathrm{m.s}^{-1})$ | [0, 0.5, 1, 2, 3, 4]      |
| $r  (\mathrm{mm})$           | [6, 12, 18, 24]           |
| h (mm)                       | [6, 12, 18, 24]           |
| s  (mm)                      | [15, 20, 25, 30]          |
| $\theta$ (°)                 | [0, 30, 60, 90]           |
| TI                           | 0.01, 0.05, 0.1, 0.2, 0.3 |

**Table 2.** Model input parameters include: inlet wind speed  $(U_{in})$ , wind attacking angle  $(\theta)$ , turbulence intensity (TI), and geometry parameters including the radius (r), height (h), and spacing (s) of the microstructures used in geometry combinations.

Finally, the most promising combinations were selected based on the maximum temperature difference within the inlet wind speed range  $(\Delta T_{U_{in,0.5}} - \Delta T_{U_{in,4}})$ .




In the second part of the study, the most promising configurations of the first part for both filled-coned and hollow-coned were modeled in 3D in a manner analogous to the two-dimensional model, with the addition of an attacking angle ranging from  $0^{\circ}$  to  $90^{\circ}$ , as specified in Table 2. Please refer to the 3D schematic shown in Fig. A1. The parametric sweep function was used to run multiple combinations of input parameters simultaneously. In addition to the variable  $U_{\rm in}$ , a variable cone radius (r), cone height (h), and spacing (s) were employed to cover 64 geometric combinations. These 64 geometric combinations were executed for all values of  $U_{\rm in}$ , and the resulting differences between forward and backward coned fibers in temperature  $(\Delta T = T_b - T_f)$ , turbulent kinetic energy  $(\Delta K = K_b - K_f)$ , and wind speed  $(\Delta U = U_b - U_f)$  on the fiber were extracted, where the subscripts b and f denote the backward and forward inlet wind, respectively. The temperature is averaged over the surface of the fiber in 2D models and over the volume in 3D models. The turbulent kinetic energy and wind speed were extracted from the surface of the fiber exposed to airflow. The first two microstructures and the three initial spacings from the inlet wind were excluded from the averaging process to exclude the instationary transition region, where the flow adjusts to the microstructures, ensuring that only fully developed flow conditions are considered. The averages were obtained using the built-in functions of the COMSOL Multiphysics model and were subsequently analyzed in Python.

The governing physics of the K- $\epsilon$  turbulence model, along with the "Heat Transfer in Solids and Fluids" module employed in this study, are described in detail in the Appendix B. Mesh independence validation is essential to balance computational efficiency with numerical accuracy (Chen et al., 2023b). To assess this, we tested four mesh sizes (Mesh A–D) with 9,272, 12,202, 14,948, and 19,834 elements, respectively. A 2D model of a hollow-coned geometry (r = 24 mm, h = 24 mm, s = 15 mm) was simulated as an example, and fiber temperatures in forward and backward-oriented FO cables were compared. Figure 3 shows the simulated temperatures for the four mesh sizes. In both orientations, differences among the meshes are minimal. The standard deviation across the meshes remains below 0.4 K at the lowest wind speed and decreases further with increasing wind speed (

Figure 1. Schematic representing the modeled fiber optic cables featuring filled-coned microstructures. Panel (a) illustrates the surface temperature, ambient air temperature, and geometric parameters, while panel (b) depicts the conceptual temperature difference between the forward and backward fibers. Here,  $T_0$  represents the temperature at the interface between the microstructure and the fiber optic cable. Forward fibers are those oriented with the wind directed toward the cone apex, whereas backward fibers are oriented with the wind opposing the cone apex.

Figure 2. Mesh distribution example of (a) filled-coned and (b) hollow-coned microstructures

**Figure 3.** Mesh independence of modeled FO cable temperatures for forward and backward fibers at different inlet wind speeds. STD denotes the standard deviation of the FO cable temperature for different mesh sizes.

**Figure 4.** Modeled temperature (T) and turbulent kinetic energy (K) in FO cables and surrounding air with filled-coned microstructures under an inlet wind speed of  $1 \text{ ms}^{-1}$ .

## 3 Results and discussion

## 3.1 Differences in modeled temperature and turbulent kinetic energy

An example of the modeled temperature (T) and turbulent kinetic energy (K) is shown in Fig. 4a and b, respectively. The results indicate that the temperature is lower for the forward fibers, with notably higher temperatures behind the cones in the backward direction. Conversely, K remains lower in the backward cones compared to the forward cones.

FO cables with hollow-coned microstructures exhibit similar behavior to those with filled-coned microstructures, with the backward direction being warmer and the forward direction cooling down (Fig. 5a and b). In filled-cone fibers, the entire cone height is attached to the fiber, whereas in hollow-cone fibers only part of the cone is attached, leaving the fiber in the

Figure 5. Modeled temperature (T) and turbulent kinetic energy (K) in and around of FO cable with attached hollow-coned microstructures and inlet wind of 1 ms<sup>-1</sup>





hollow section exposed to airflow. This reduced PVC attachment in hollow-cone fibers results in less heat transfer within the microstructure itself, so the fiber temperature is less affected by the surrounding material. A longer exposed fiber length additionally results in a shorter thermal response time. In the present study, however, the model was run only until steady-state heat transfer was reached, and the response time was not examined in detail. In contrast, the more PVC material printed on filled-cone fibers, the longer it takes for the fiber temperature to equilibrate with the air temperature, due to PVC's lower thermal conductivity compared to copper or steel. The modeling results confirmed the viability of the fundamental principle behind the microstructure approach in creating a temperature difference between the forward and backward FO cables, which is induced by variations in directional heat loss. The relationship between the temperature difference ( $\Delta T$ ) and the inlet wind speed ( $U_{in}$ ) in both microstructure types shown in 6a and b aligns with the findings of Lapo et al. (2020), where  $\Delta T$  is higher at low wind speeds and decreases non-linearly with increasing inlet wind speed. The comparison of the best-performing microstructure design (r = 6 mm, h = 12 mm, s = 20 mm) reported by Lapo et al. (2020) with the modeled combination illustrated in Fig. 6a shows a  $R^2$  of 0.95, indicating that the model behaves similarly to the wind tunnel experiment (The comparison graph is shown in Appendix A4.). Furthermore, in this study, the most promising combinations were selected based on the maximum  $(\Delta T_{U_{in.0.5}} - \Delta T_{U_{in.4}})$ , which equals 1.15 K in Lapo et al. (2020) and 0.98 K in the model. However, the magnitude of the modeled  $\Delta T$  in this study differs from the wind tunnel  $\Delta T$  by an offset of 1.32 K. The following points may explain this discrepancy:

- The simplifications applied to the FO cable in the model. In the wind tunnel experiment, the FO cable consisted of a thin metal sheath filled with jelly and glass cores, while in the model it was simplified as a solid copper bar.

- Differences in turbulence characteristics between the wind tunnel and the COMSOL model. In wind tunnels, turbulence intensity and scales arise from inflow generation and boundary layer development, whereas in COMSOL the K- $\epsilon$  closure with wall functions represents turbulence in an averaged, isotropic way. This difference can reduce the fidelity of convective heat transfer predictions, particularly around small-scale cable structures.


- The difference in FO cable materials also leads to differences in heating rate between the wind tunnel and the model, which can affect the magnitude of  $\Delta T$ .

Based on the 2D modeling, the four most promising geometric combinations for filled-coned and hollow-coned FO cables were depicted in Fig. 6a and b, which were selected based on the  $\Delta T$  magnitude between the  $U_{in}$  = 0.5 and 4 ms<sup>-1</sup>, plotted as marked bold lines. For example, the combination of hollow-coned microstructure (radius = 24 mm, height = 24 mm, and spacing = 15 mm) produced a temperature difference ( $\Delta T_{U_{in,0.5}}$  -  $\Delta T_{U_{in,4}}$ ) of 4.78 K. The geometry used in the previous study is also added to the plot as a reference. We used  $\Delta T$  to identify the most promising cone geometry combinations; for completeness, the combinations selected based on  $\Delta K$  and  $\Delta U$  are also provided in Appendix Figures A2 and A3. The filledconed microstructures are able to induce a temperature difference of 3.57 K at 0.5 ms<sup>-1</sup> and a  $(\Delta T_{U_{in,0.5}} - \Delta T_{U_{in,4}})$  of 2.61 K between 0.5 and 4 ms<sup>-1</sup> with the geometry parameters of r = 18 mm, h = 24 mm, and s = 15 mm where the hollow-coned combinations reached 7.25 K at 0.5 ms<sup>-1</sup> and a ( $\Delta T_{U_{in,0.5}}$  -  $\Delta T_{U_{in,4}}$ ) of 4.78 K between 0.5 and 4 ms<sup>-1</sup> with the geometry parameters of r=24 mm, h=24 mm, and s=15 mm (Fig. 6a and b). Both microstructure types show improvement in  $\Delta T$ magnitude compared to (Lapo et al., 2020). Considering the flow conditions over the FO cables, in filled-coned microstructures, a decrease in turbulent kinetic energy and total wind speed (U) is determined down to  $-0.21 \text{ m}^2\text{s}^{-2}$  and  $-0.55 \text{ ms}^{-1}$ , respectively, 220 where the top  $\Delta T$  combinations are not necessarily the ones with the lowest  $\Delta K$  and  $\Delta U$  (Fig. 6c and e). The reason is part of the FO cable is covered with PVC microstructures and the heat exchange is a function of the turbulent heat loss as well as the heat transfer within the microstructures. In contrast to filled-coned, the highest  $\Delta T$  combinations also show a lower  $\Delta K$  and  $\Delta U$  of -0.11 m<sup>2</sup>s<sup>-2</sup> and -0.21 ms<sup>-1</sup> in hollow-coned combinations, where a smaller part of the FO cable is covered with microstructures (Fig. 6d and f). Our findings indicate that aspect ratio alone is not a significant geometric factor in influencing the temperature difference on the FO cable, as configurations with similar aspect ratios exhibit different temperature differences and the thermal properties of the material used in the FO cable, as well as its volume, influence the thermal heat loss from cable. To select the optimal microstructures to implement in observational experiments, additional criteria beyond the highest temperature difference must be considered. The  $\Delta T$  should be sufficiently large to be detectable by FODS as the 230 standard uncertainty for the conventional FO device, Silixa Ultima-S DTS system (Hertfordshire, UK), is defined as 0.36 C° for sampling every second, over every 12.7 cm and for FO cables less than 500 meters (des Tombe et al., 2020). In addition, the microstructures should cover as little of the FO cable as possible with PVC, minimizing any increase in the fiber's response time to temperature changes and reducing the extra weight associated with additional costs, Based on before mentioned criterion, the geometry combinations of r=18 mm, h=24, and s=20 mm in filled-coned and r=24 mm, h=24, and s=15 mm in hollow-coned is selected as the most promising combinations to further investigate the effect of turbulence intensity on resulted 235  $\Delta T$ 's.

Figure 6. The differences in modeled temperature ( $\Delta T$ ) (a, b), turbulent kinetic energy ( $\Delta K$ ) (c, d), and total wind speed ( $\Delta U$ ) (e, f) between forward and backward fiber optic cables are presented for varying inlet wind speeds in filled-coned and hollow-coned microstructures, respectively, across 64 geometric combinations. The four combinations producing the highest  $\Delta T$ , along with one corresponding to the geometry of the filled-coned microstructure modeled in (Lapo et al., 2020), are shown as solid lines with markers, while the remaining combinations are depicted as faded lines.

# 3.2 Effects of turbulence intensity on $\Delta T$ and $\Delta k$



The dynamic nature of the atmospheric boundary layer generates varying flow conditions near the Earth's surface, resulting in a range of turbulence intensities. To evaluate the performance of the selected microstructure configurations under different turbulence conditions, we applied a conventional range of TI values (Table 2) to the inflow of the 2D model. The filled-coned configuration demonstrated the capability to maintain a  $\Delta T > 1$  K for TI values below 0.2 and inlet velocities ( $U_{in}$ ) up to 2 ms<sup>-1</sup>, as illustrated in Figure 7a. This temperature difference is significantly large to be detected using FODS. Furthermore, the magnitude of the  $\Delta K$  on the FO cable exhibited a higher sensitivity to TI compared to the inlet wind speed. This observation underscores the critical role of TI in governing directional heat loss from the FO cable. On the other hand, the hollow-coned microstructure maintained a  $\Delta T$  exceeding 2 K across all turbulence intensities and for inlet wind speeds of up to 4 ms<sup>-1</sup> (Fig. 8a). The  $\Delta K$  decreased with increasing inlet wind speed across the range of TI values, exhibiting only a minimal increase

Figure 7. (a) The variation of (a)  $\Delta T$  and (b)  $\Delta K$  across different turbulence intensities and inlet wind speeds for the selected filled-coned microstructure based on 2D model. This map and also the map in Fig. 8 are generated using cubic interpolation of discrete values of the modeled  $\Delta T$  and  $\Delta K$  across different TI's and inlet wind speeds.

in TI between 0.1 and 0.15 (Fig. 8a). These findings indicate that the selected hollow-coned microstructure outperforms the filled-coned configuration across the range of tested turbulence intensities.

# 3.3 Wind attack angle effect on $\Delta T$ and $\Delta k$

The wind attack angle is critically important for real-world applications at the field scale, since the turbulent flow varies substantially in speed and direction. The inclusion of hollow-coned microstructures in this study aimed at decreasing the sensitivity of the FO cable's temperature difference (ΔT) to lateral winds, i.e. flow component orthogonal to the orientation of the FO cable, when compared to the previous study. Consequently, the wind attack angle effect was analyzed exclusively for FO cables featuring hollow-coned microstructures with a 3D model. In the analysis, inlet wind was applied at attack angles of 0°, 30°, 60°, and 90° relative to the fiber. The 0° attack angle corresponds to wind aligned along the fiber, whereas the 90° angle represents wind orthogonal to the fiber. As expected, the hollow-coned configuration with geometry parameters of r = 24 mm, h = 24, and s = 15 exhibited a ΔT of up to 0.8 K at a wind attack angle of 60° (Figure 9a). Furthermore, this configuration demonstrated a distinct decay pattern in ΔT across various wind speeds and attack angles, a crucial characteristic for analyzing real FO cables. The turbulent kinetic energy difference (ΔK) increased with inlet wind speed, reaching its maximum at a wind attack angle of 30°, where the maximum ΔK was observed to be -0.32 m<sup>2</sup>s<sup>-2</sup> (Figure 9b).

Figure 8. (a) The variation of (a)  $\Delta T$  and (b)  $\Delta K$  across different turbulence intensities and inlet wind speeds for the selected hollow-coned microstructure based on 3D model.

Figure 9. Directional sensitivity of  $\Delta T$  for the selected geometric configuration of hollow-cone microstructures. This map is generated using cubic interpolation of discrete values of the modeled  $\Delta T$  and  $\Delta K$  across different wind attack angles and inlet wind speeds.

## 4 Conclusions







This study investigated the influence of varying geometry of microstructures attached to FO cables on the thermal and turbulence responses under varying flow conditions and turbulence intensities. A comparative evaluation of geometric configurations identified specific combinations of hollow-cone- and filled-coned microstructures which optimize  $\Delta T$  while balancing additional criteria, such as maintaining sensitivity to wind speed and minimizing PVC coverage on the FO cable. The findings highlight the potential to improving upon earlier numerical and experimental studies employing the same principle by optimizing geometry of the microstructures including their radius, height, spacing, shape, and hollow or filled while at the same time reducing the sensitivity of the heated filled-coned FO cable to flow components orthogonal to the cable. The results demonstrated that FO cables with filled-coned microstructures with a radius of 18 mm, height of 24, and spacing of 20 mm outperformed the earlier designs. Moreover, FO cables with hollow-coned microstructures, with parameters r = 24 mm, h=24 mm, and s=15 mm, exhibited superior performance over the tested conditions. These hollow-coned configurations maintained  $\Delta T$  values exceeding 2 K across all turbulence intensities and wind speeds up to 4 m s<sup>-1</sup>, demonstrating robustness across a wide range of flow conditions. In contrast to the study by Lapo et al. (2020), which primarily considers the aspect ratio and spacing of the cones to determine the optimal configuration, our findings demonstrate that aspect ratio alone is not a decisive geometric factor in influencing the temperature difference on the FO cable. Instead, the cone's overall dimensions play a crucial role, as configurations with similar aspect ratios exhibit different temperature differences. This study also emphasizes the importance of integrating heat transfer and turbulent flow, considering not only turbulence but also the thermal properties and volume of the material, which are directly linked to cone dimensions. Sensitivity analysis of wind attack angles revealed that the hollow-coned combination achieved a  $\Delta T$  of up to 0.8 K at a 60° angle, with a distinct decay pattern across various wind speeds. This analysis is important because in the real-world, 3D turbulent flows are inherently stochastic in both wind speed and direction. This behavior underscores the significance of microstructure design in mitigating the impact of orthogonal wind and preserving temperature gradients along the FO cable. These results pave the way for deploying optimized FO cable designs specifically for weak-wind boundary layer studies, enhancing their capability to accurately capture wind direction and observe distributed turbulent heat fluxes using FODS as demonstrated in Abdoli et al. (2023).

Future research should prioritize experimental validation of the proposed designs under field conditions to confirm their robustness and reliability in real-world applications. Such studies will further advance the applicability of FO cables in atmospheric boundary layer research, particularly in enhancing the accuracy of distributed heat flux and turbulence measurements.

Data availability. The datasets generated and analyzed during the current study are publicly available in the Zenodo repository under the following link: https://doi.org/10.5281/zenodo.17136833

#### Appendix A

Figure A1. A 3-dimensional schematic of the hollow-coned model with inlet wind vectors varying with  $\theta$ . The light blue surfaces were designated as inlets to expose the FO cables to constant wind speeds at different angles.

Figure A2. The differences in temperature ( $\Delta T$ ) (a, b), turbulent kinetic energy ( $\Delta K$ ) (c, d), and total wind speed ( $\Delta U$ ) (e, f) between forward and backward fiber optic cables are presented for varying inlet wind speeds in filled-coned and hollow-coned microstructures, respectively, across 64 geometric combinations. The four combinations producing the highest  $\Delta K$ , along with one corresponding to the geometry of the filled-coned microstructure modeled in Lapo et al. (2020), are shown as solid lines with markers, while the remaining combinations are depicted as faded lines.

Figure A3. The differences in temperature ( $\Delta T$ ) (a, b), turbulent kinetic energy ( $\Delta K$ ) (c, d), and total wind speed ( $\Delta U$ ) (e, f) between forward and backward fiber optic cables are presented for varying inlet wind speeds in filled-coned and hollow-coned microstructures, respectively, across 64 geometric combinations. The four combinations producing the highest  $\Delta U$ , along with one corresponding to the geometry of the filled-coned microstructure modeled in (Lapo et al., 2020), are shown as solid lines with markers, while the remaining combinations are depicted as faded lines.

# Appendix B


## **B1** Governing physics

Turbulence was modeled using the  $K-\epsilon$  approach, and the "Heat Transfer in Solids and Fluids" module was employed to simulate the thermal interactions between the filled-coned fiber and the ambient air. The bidirectional coupling between turbulent flow and heat transfer was achieved by adopting the non-isothermal flow coupling interface. To obtain more accurate solution results, the Kays-Crawford heat transfer turbulence model was adopted in the simulation studies. The following sections present the governing equations of the employed model, including those pertaining to heat transfer in solids and fluids, as well as turbulent flow.

Figure A4. The comparison of the best performing filled-cone microstructure design (r = 6 mm, h = 12 mm, s = 20 mm) reported by Lapo et al. (2020) with the one modeled in this study.

#### **B1.1** Heat Transfer in Solids

The heat conduction in a solid is described by the Fourier equation, where heat transfers from the higher temperature point to the lower temperature point. The following equation describes the heat transfer in solids:

$$\rho_s C_p \left( \frac{\partial T}{\partial t} + \mathbf{u}_{trans} \cdot \nabla T \right) + \nabla \cdot (\mathbf{q} + \mathbf{q}_r) = -\alpha : \frac{dS}{dt} + Q_{add}$$
(B1)

Where  $\rho_s$  represents the density of the solid,  $C_p$  denotes the specific heat at constant pressure, and T stands for the absolute temperature. The velocity vector of translational motion is given by  $\mathbf{u}_{trans}$ , while  $\mathbf{q}$  and  $\mathbf{q}_r$  correspond to the heat flux by conduction and by radiation, respectively. The coefficient of thermal expansion is denoted by  $\alpha$ , and S is the second Piola-Kirchhoff stress tensor. Finally,  $Q_{add}$  accounts for any additional heat sources present in the system. The colon (:) represents the double dot product (or tensor contraction) between two second-order tensors.

## **B1.2** Heat Transfer in Fluids

The heat transfer in fluids can be described by the following equation:

$$\rho C_p \left( \frac{\partial T}{\partial t} + \mathbf{u} \cdot \nabla T \right) + \nabla \cdot (\mathbf{q} + \mathbf{q}_r) = Q_p + Q_v + Q_{add}$$
 (B2)

where:

305

$$Q_p = \alpha_p T \left( \frac{\partial p}{\partial t} + \mathbf{u} \cdot \nabla p \right), \quad Q_v = \tau : \nabla \mathbf{u}$$
(B3)

Where p denotes the pressure, and  $\alpha_p$  is the coefficient of thermal expansion. The velocity vector is given by  $\mathbf{u}$ , while  $\mathbf{q}$  and  $\mathbf{q}_r$  represent the heat flux by conduction and radiation, respectively. The viscous stress tensor is denoted by  $\tau$ . The coefficient

of thermal expansion  $\alpha_p$  is given by:

$$\alpha_p = -\frac{1}{\rho} \frac{\partial \rho}{\partial T} \tag{B4}$$

In accordance with the simulation's stationary mode, the time-dependent variables presented in equations (1–4) are equal to zero.

#### **B1.3** Turbulent Flow

The  $k - \epsilon$  model contains two dependent variables: turbulent kinetic energy (K) and the turbulent dissipation rate  $(\epsilon)$ . The eddy viscosity  $(\mu_T)$  is defined as:

$$\mu_T = \rho C_\mu \frac{k^2}{\epsilon} \tag{B5}$$

and the transport equation for K is as follows:

$$\rho \frac{\partial k}{\partial t} + \rho \mathbf{u} \cdot \nabla k = \nabla \cdot \left[ \left( \mu + \frac{\mu_T}{\sigma_k} \right) \nabla k \right] + P_k - \rho \epsilon \tag{B6}$$

where the production term  $P_k$  is:

$$P_k = \mu_T \left[ \nabla \mathbf{u} : \left( \nabla \mathbf{u} + (\nabla \mathbf{u})^T \right) - \frac{2}{3} (\nabla \cdot \mathbf{u})^2 \right] - \frac{2}{3} \rho k \nabla \cdot \mathbf{u}$$
(B7)

The transport equation for  $\epsilon$  is as follows:

$$\rho \frac{\partial \epsilon}{\partial t} + \rho \mathbf{u} \cdot \nabla \epsilon = \nabla \cdot \left[ \left( \mu + \frac{\mu_T}{\sigma_{\epsilon}} \right) \nabla \epsilon \right] + C_{\epsilon 1} \frac{\epsilon}{k} P_k - C_{\epsilon 2} \rho \frac{\epsilon^2}{k}$$
(B8)

Where the constant values are determined as follows:  $C_{\mu} = 0.09$ ,  $C_{\epsilon 1} = 1.44$ ,  $C_{\epsilon 2} = 1.92$ ,  $\sigma_k = 1.0$ , and  $\sigma_{\epsilon} = 1.3$ .

*Author contributions.* Conceptualization: MA; methodology and manufacturing: MA and RP; formal analysis: MA; investigation: MA; data curation: MA; writing – original draft preparation: MA; writing – review and editing: MA and CKT; visualization: MA; project administration: CKT and MA; funding acquisition: CKT. All authors have read and agreed to the published version of the paper.

Competing interests. The authors declare that they have no competing interests.

Acknowledgements. This research was supported by the European Research Council, H2020 European Research Council (DarkMix (grant 335 no. 724629)).

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
