# Peer review of "Improving turbulent airflow direction measurements for fiber-optic distributed sensing using numerical simulations"

_EGUsphere, 2025_

## Author Comment (AC1)

**Response to Reviewer Comments**

Manuscript Title: Improving turbulent airflow direction measurements for fiber-optic distributed sensing using numerical simulations

Authors: Mohammad Abdoli, Reza Pirkhoshghiyafeh, and Christoph K. Thomas

Date: 2025.08.30

Dear Reviewer,
Thank you very much for your constructive general and technical comments. We are confident that your feedback has significantly improved the quality and clarity of our work. We carefully considered each of your suggestions and revised the manuscript accordingly. In the following, we provide a point-by-point response outlining how your comments were addressed, including additional clarifications, figures, and expanded discussions where necessary. We believe these changes strengthen the scientific contribution and presentation of the paper.

The all line numbers referenced here is based on the revised version.

**Overall Comments:**

**Comment 1.**

The introduction provides only a brief overview of the "microstructure approach". To better frame the study's contribution, the authors should expand on the underlying physics. Specifically, the text should clarify why maximizing the temperature difference between the cables is the primary optimization goal and explicitly describe the roles that turbulence kinetic energy and mean velocity differences play in the technique. A clearer understanding of the physical principles is essential for appreciating the study's objectives.

Response: The introduction has been updated by expanding the description of the underlying physics of the microstructure approach (Lines 58–69). In particular, we now discuss how changes in microstructure orientation influence turbulence intensity and mixing around the fiber-optic cables, where the role of TKE is emphasized. In response to Comment 21, we have also emphasized the role of mean flow velocity. Furthermore, we clarified within the manuscript why maximizing the temperature difference is a key aspect of this approach (Lines 94-97).

**Comment 2.**

The study must better demonstrate the validity and quality of its numerical simulations.

Response: The quality of the numerical simulation was evaluated through a mesh independence test. A representative geometry configuration was simulated using four different mesh sizes, and the resulting fiber temperatures were compared for both forwardand backward-oriented fibers. Across all cases, the standard deviation between meshes remained below 0.4 °C at the lowest wind speed and decreased further with increasing wind speed, reaching values below 0.1 °C at $U_{in}$=4 ms$^{-1}$. This analysis has been included at the end of the Methods section (Lines 159-166).

First, the authors should add appropriate references that validate the use of this specific COMSOL setup and its turbulence models for similar heat transfer simulations.

Response: We have revised the Methods text to include supporting citations demonstrating the suitability of our COMSOL configuration for coupled heat transfer and fluid–solid interactions (Lines 103-108).

Second, the comparison with the experimental results from Lapo et al. (2000) needs to be presented in details. Just mentioning that "the magnitude of the modeled ΔT in this study differs from the ΔT observed by Lapo et al. (2020) in the wind tunnel experiment" is insufficient. Furthermore, the potential reasons for the discrepancy mentioned are too generic and broad, it does not clarify if the model can be trusted. The authors should present a clear argument for why the simulation results remain valid and useful for the design optimization process, even with this difference.

Response: A comparison was performed with the results of Lapo et al. (2020), and the outcome is presented in lines 194-208, with a corresponding comparison graph added to the appendix. Potential reasons for discrepancies were discussed in detail.

**Specific Comments:**

**Comment 1.**
l. 58: "field experiment in the field" improve

Response: Done!

**Comment 2.**
l. 80: "Our primary objective is to improve the temperature difference between the forward (wind directed towards the cone apex) and backward (wind opposite to the cone apex) filled-coned fibers" why exactly?

Response: The discussion is included in (Lines 92-95). "A larger temperature difference between the forward and backward fibers ensures that the signal exceeds the reported measurement erorr 0 to 0.1K reported by (Schilperoort et al., 2020). It therefore enhances the precision of the microstructure approach at higher wind speed ranges and increasing the persistence of the temperature difference under lateral wind condition"

**Comment 3.**
l. 89: 16r (use mathematical notation)

Response: Done!

**Comment 4.**

l. 88-90: "determined" how? Or just chosen?

Response: The dimensions were determined. "The dimensions were selected and optimized with several test runs to maintain accuracy and reduce computational costs."

**Comment 5.**

l. 102 : COMSOl

Response: Done!

**Comment 6.**

l. 106 (Lapo et al., 2022)

Response: Edited.

**Comment 7.**

l. 112: how do you know it is resolving the boundary layers? Can you show it? Is it really necessary?

Response: We used the standard K-ε turbulence model with wall-function treatment in COMSOL. In this approach, the near-wall viscous sublayer is not explicitly resolved; instead, the boundary layer is modeled using wall functions. This treatment is widely used for engineering-scale turbulent heat transfer and is appropriate for the quantities of interest in this study (bulk temperature distribution and convective heat flux).

The statement edited to "Additional adjustments were applied to the mesh manually to ensure adequate resolution of the boundary-layer profiles and the microstructure edges."

**Comment 8.**

l. 112: designed

Response: Done!

**Comment 9.**

l. 120: "in both forward and backward wind flows", what do you mean?

Response: The sentenced was changed to „ These 64 geometric combinations were executed for all values of $U_{in}$, and the resulting differences between forward and backward coned fibers in temperature ($\Delta T$) and turbulent kinetic energy ($\Delta k$) on the fiber were extracted."

**Comment 10.**

l. 120: "resulting differences in temperature ($\Delta T$) and turbulent kinetic energy ($\Delta k$) on the fiber were extracted." it is not clear which difference is it here, difference between the fibers? Calculated where? I think it is partially explained in Fig. 1b and later in the text, but it should be better explained here.

Response: The "Data analysis" section moved to this paragraph (Line 155) including the $\Delta T$ $\Delta k$, $\Delta U$ and how and where they were averaged.

"These 64 geometric combinations were executed for all values of Uin , and the resulting differences between forward and backward coned fibers in temperature ($\Delta T = T_b - T_f$ ), turbulent kinetic energy ($\Delta K = K_b - K_f$ ), and wind speed ($\Delta U = U_b - U_f$ ) on the fiber were extracted, where the subscripts b and f denote the backward and forward inlet wind, respectively. The temperature is averaged over the surface of the fiber in 2D models and over the volume in 3D models. The turbulent kinetic energy and wind speed were extracted from the surface of the fiber exposed to airflow "

**Comment 11.**
In the text, separate discussion of fig 1 between fig 1a and fig 1b, and I think that fig 1b was not mentioned.

Response: Added.

**Comment 12.**
Fig1a: define who is the forward/backward fiber

Response: Added to subtitle.

**Comment 13.**
Fig 1: "Here, T0 represents the temperature at the interface between the microstructure and the fiber optic cable." T0 is very small in the figure, it took me a while to find it...

Response: $T_0$ reformatted.

**Comment 14.**
Fig 1: Why is the temperature in the upper fiber much lower than the temperature in the lower fiber and in the cones?

Response: This temperature difference arises from the cone orientation. In forward coned fibers (aligned with the wind), the flow penetrates the cone openings and enhances cooling of the fiber. In contrast, in backward coned fibers, the cone acts as a shelter that reduces exposure to the flow and thereby lowers the cooling rate. This mechanism underpins the microstructure approach, which generates a directional temperature difference between forward and backward coned fibers.

**Comment 15.**
l. 121: why is the value of $\Delta k$ important to monitor?

Response: As also discussed in general comments, changes in microstructure orientation influence turbulence intensity and mixing around the fiber-optic cables which directly influence the heat loss from the fiber.

**Comment 16.**

l. 130: instationary

Response: Done!

**Comment 17.**

l. 139: "The hollow-coned design has less PVC attaching length to the fiber, resulting in less heat transfer within the microstructure itself, which has less of an impact on the FO temperature compared to the filled cone cables." how did you come to this conclusion?

Response: In filled-cone fibers, the entire cone height is attached to the fiber, whereas in hollow-cone fibers only part of the cone is attached, leaving the fiber in the hollow section exposed to airflow. A greater exposed fiber length can result in a shorter thermal response time. Conversely, the more PVC material printed on the fiber, the longer it takes for the fiber temperature to equilibrate with the air temperature, due to PVC's lower thermal conductivity compared to copper or steel. The explanation added to the text (Lines 182-190).

**Comment 18.**

increase the size of figs 3 and 4

Response: Done!

**Comment 19.**

fig 3 and 4: TKE is much much higher in hollow case, why?

Response: As noted in the general comments, the turbulence kinetic energy (TKE) is much higher in the hollow case because the geometry of the hollow cone increases surface roughness compared to the filled cone, thereby enhancing turbulence around the hollow cones.

**Comment 20.**

l. 144: "The relationship between the temperature difference ($\Delta T$ ) and the inlet wind speed (U in ) aligns with the findings of Lapo et al. (2020), where $\Delta T$ is higher at low wind speeds and decreases non-linearly with increasing inlet wind speed." Is this conclusion from Fig 5? If so, say it explicitly.

Response: Done.

**Comment 21.**

How was $\Delta U$ quantified? Why is it important?

Response: The $\Delta U$ is the wind speed difference that reaches the fiber surface. The text updated to include the $\Delta U$ (Line 161). It was important to include the $\Delta U$ since, in the heat transfer equation (Appendix B8), both the turbulent kinetic energy and the velocity vector appear as driving terms; while $\Delta K$ quantifies turbulence-related effects, $\Delta U$ represents the direct convective transport potential. Because conjugate turbulent heat transfer results

from the interplay between conduction within the fiber and convection and turbulence by the surrounding airflow, variations in $\Delta U$ indicate how strongly the fiber is exposed to asymmetric flow conditions, which directly influence surface heat exchange. Being aware of the critical influence of turbulent kinetic energy in heat exchange between the fibers and the surrounding air particularly between the cone spacings, we included $\Delta U$ as a complementary variable to give idea about the role of velocity asymmetry in the directional cooling sensitivity of the fiber.

**Comment 22.**
Fig 5: I don't understand how cases producing the highest $\Delta T$ ware chosen, highest for which velocity? From Fig 5b they are the highest, but in Fig 5a they are not necessarily, right?

Response: The criteria is based on the maximum temperature difference between ($\Delta T_{Uin = 0.5}$ - $\Delta T_{Uin = 4}$ ), showing that which is mentioned in the text (Line 154), The equation is added to the text for more clarification.

**Comment 23.**
l. 144: "The relationship between the temperature difference ($\Delta T$) and the inlet wind speed ($Uin$) aligns with the findings of Lapo et al. (2020), where $\Delta T$ is higher at low wind speeds and decreases non-linearly with increasing inlet wind speed" Is this statement based on Fig 5? If so, say it explicitly.

Response: The statement is based on Fig. 5, the text is edited.

**Comment 24.**
l. 152: "the selected combinations illustrated also based on $\Delta k$ and $\Delta U$ shown in appendix Fig. A2 and A3" improve .

Response: Improved to "We used $\Delta T$ to identify the most promising cone geometry combinations; for completeness, the combinations selected based on $\Delta K$ and $\Delta U$ are also provided in Appendix Figures A2 and A3."

**Comment 25.**
l. 156: compared to Lapo et al. (2020)

Response: Done!

**Comment 26.**
l. 167: "implement in observational experiments, additional criteria"

Response: Done!

**Comment 27.**
Secs 3.2 and 3.3, figs 6 and 7: are these results based on 3D simulations? If so, state it explicitly.

Response: Section 3.2 is based on 2D simulations, and section 3.3 is based on 3D simulations. Explanations were added to methods, sections 3.2 and 3.3, as well as to the figure subtitles.

**Comment 28.**
Fig A2: modeled in Lapo et al. (2020)

Response: Done!

---

## Author Comment (AC2)

**Response to Reviewer Comments**

Manuscript Title: Improving turbulent airflow direction measurements for fiber-optic distributed sensing using numerical simulations

Authors: Mohammad Abdoli, Reza Pirkhoshghiyafeh, and Christoph K. Thomas

Date: 2025.08.30

**General Comments:**

This manuscript presents a comprehensive numerical investigation of microstructure geometries for fiber-optic (FO) cables used in turbulent airflow measurements. The research addresses an important gap in fiber-optic distributed sensing (FODS) regarding the effects of microstructures' geometry used in FO cable on dynamic parameters important for atmospheric boundary layer studies. The study builds upon previous work by Lapo et al. (2020), and extends the analysis to include hollow-cone microstructures while examining a broader range of geometric parameters and airflow direction effects.

The introduction of hollow-cone microstructures represents a meaningful innovation that demonstrates superior performance compared to filled-cone designs. The authors systematically investigated 64 geometric combinations, providing a thorough exploration of design parameters. In addition, the authors incorporated appropriate real-world factors, such as turbulence intensity variations and wind attack angles, and considered the detection limits of commercial FODS systems.

The study is well-written, providing practical guidance for future FODS implementation while clearly stating the main limitations and highlighting the need for future field/lab experiments. These findings establish a foundation for more accurate wind direction measurements, distributed turbulent heat flux assessments, and the detection of vertical wind speed perturbations using FODS, representing substantial progress toward enhanced atmospheric monitoring capabilities.

Dear Dr. Rosalem,

Thank you very much for your positive and constructive general comments. We sincerely appreciate your recognition of our work and its contribution to advancing fiber-optic distributed sensing for atmospheric applications. We have carefully considered all your general and technical comments and have addressed them thoroughly in the revised manuscript.

The all line numbers referenced here is based on the revised version.

**Specific Comments:**

**Comment 1.**

The computational approach is well-described with appropriate boundary conditions and mesh considerations. However, the authors should consider adding some references that applied the k-ε model and commenting on the model's limitations and constraints. Additionally, it would be beneficial (if possible) to include information about uncertainty quantification or confidence intervals for the computed results.

References applying the k-ε model have been added (Lines 121–129), and the limitations and constraints of the model are now discussed. In addition, we assessed the quality of the numerical simulations through a mesh independence test. A representative geometry configuration was simulated using four different mesh sizes, and the resulting fiber temperatures were compared for both forward- and backward-oriented fibers. Across all cases, the standard deviation between meshes remained below 0.4 °C at the lowest wind speed and decreased further with increasing wind speed, reaching values below 0.1 °C at 4 ms$^{-1}$ (Lines 174-176)

**Technical Comments:**

**Comment 1.**
Line 74: "In this numerical study, we aim to overcome the major weaknesses of the existing microstructure approach in order to enhance wind speed and direction measurements using filled-coned FO cables while minimizing their sensitivity to lateral wind flows."

Response: It is not entirely clear what is meant by this comment. Could the reviewer please clarify what specific aspect should be demonstrated or elaborated on? This will help us address the concern more accurately in the revised manuscript.

**Comment 2.**
Line 108: Is it "at lower wind speeds" right?

Response: It is actually for both higher and lower wind speeds. Here we discussed the magnitude of temperature difference between the electrically heated fiber optic cables in comparison with air temperature. It should be large enough to maintain a positive temperature difference within the wind speed range of the environment (in this modeling 0-4 ms$^{-1}$).

The text edited to "In addition, the temperature difference between air and FO cables was chosen to be large enough to prevent excessive cooling at wind speed range of the modeling."

**Comment 3.**
Line 160: "Similarly, ..."

Response: The "In contrast .." is true here, since we discuss the contrast of behavior of filled-coned and hollow-coned temperature difference. The combinations with higher K and ΔU in filled-coned does not mean necessarily the highest ΔT combinations, but in hollow-coned does.

We edited "In contrast,…" to "In contrast to filled-coned, "

**Comment 4.**

References: Please review this section to ensure consistency in the formatting of the reference list.

Response: Done!